# β-Catenin in Dendritic Cells Negatively Regulates CD8 T Cell Immune Responses through the Immune Checkpoint Molecule Tim-3

**DOI:** 10.3390/vaccines12050460

**Published:** 2024-04-25

**Authors:** Chunmei Fu, Jie Wang, Tianle Ma, Congcong Yin, Li Zhou, Björn E. Clausen, Qing-Sheng Mi, Aimin Jiang

**Affiliations:** 1Center for Cutaneous Biology and Immunology, Department of Dermatology, Henry Ford Health, Detroit, MI 48202, USA; cfu1@hfhs.org (C.F.); jwang5@hfhs.org (J.W.); cyin1@hfhs.org (C.Y.); lzhou1@hfhs.org (L.Z.); 2Immunology Program, Henry Ford Cancer Institute, Henry Ford Health, Detroit, MI 48202, USA; 3College of Human Medicine, Michigan State University, East Lansing, MI 48824, USA; 4Department of Computer Science and Engineering, School of Engineering and Computer Science, Oakland University, Rochester, MI 48309, USA; tianlema@oakland.edu; 5Department of Internal Medicine, Henry Ford Health, Detroit, MI 48202, USA; 6Institute for Molecular Medicine, Paul Klein Center for Immune Intervention, University Medical Center of the Johannes Gutenberg-University Mainz, Langenbeckstrasse 1, 55131 Mainz, Germany; bclausen@uni-mainz.de

**Keywords:** dendritic cell-based vaccines, β-catenin, Tim-3, immune checkpoint blockade, CD8 T cell immunity

## Abstract

Recent studies have demonstrated that β-catenin in dendritic cells (DCs) serves as a key mediator in promoting both CD4 and CD8 T cell tolerance, although the mechanisms underlying how β-catenin exerts its functions remain incompletely understood. Here, we report that activation of β-catenin leads to the up-regulation of inhibitory molecule T-cell immunoglobulin and mucin domain 3 (Tim-3) in type 1 conventional DCs (cDC1s). Using a cDC1-targeted vaccine model with anti-DEC-205 engineered to express the melanoma antigen human gp100 (anti-DEC-205-hgp100), we demonstrated that CD11c-β-catenin^active^ mice exhibited impaired cross-priming and memory responses of gp100-specific CD8 T (Pmel-1) cells upon immunization with anti-DEC-205-hgp100. Single-cell RNA sequencing (scRNA-seq) analysis revealed that β-catenin in DCs negatively regulated transcription programs for effector function and proliferation of primed Pmel-1 cells, correlating with suppressed CD8 T cell immunity in CD11c-β-catenin^active^ mice. Further experiments showed that treating CD11c-β-catenin^active^ mice with an anti-Tim-3 antibody upon anti-DEC-205-hgp100 vaccination led to restored cross-priming and memory responses of gp100-specific CD8 T cells, suggesting that anti-Tim-3 treatment likely synergizes with DC vaccines to improve their efficacy. Indeed, treating B16F10-bearing mice with DC vaccines using anti-DEC-205-hgp100 in combination with anti-Tim-3 treatment resulted in significantly reduced tumor growth compared with treatment with the DC vaccine alone. Taken together, we identified the β-catenin/Tim-3 axis as a potentially novel mechanism to inhibit anti-tumor CD8 T cell immunity and that combination immunotherapy of a DC-targeted vaccine with anti-Tim-3 treatment leads to improved anti-tumor efficacy.

## 1. Significance

Recent studies have shown that tumor-induced β-catenin activation in DCs suppresses CD8 T cell immunity by inhibiting cross-priming, although the underlying mechanisms remain incompletely understood. In this report, we identified the immune checkpoint molecule Tim-3 as a potential downstream target upregulated by β-catenin in DCs and uncovered the β-catenin/Tim-3 axis as a new mechanism to inhibit DC vaccine-induced CD8 T cell responses. Based on these new findings, we further demonstrated that the combination of DC-targeted vaccines and anti-Tim-3 immune checkpoint blockade (ICB) improves the anti-tumor efficacy of DC vaccines. 

## 2. Introduction

As the initiators of adaptive immune responses, dendritic cells (DCs) are known as the most potent antigen-presenting cells (APCs) to cross-present tumor-associated antigens (TAAs) and prime tumor antigen-specific CD8 T cells (termed cross-priming) to control tumor growth [1,2]. This unique functionality of DCs makes DC-based vaccines, which aim to potentiate host anti-tumor immunity, especially CD8 T cell immunity, one of the leading strategies for cancer immunotherapy [3,4]. Current clinical trials with DC-based vaccines, however, have shown limited success [3,4]. As DC-based vaccines generally rely on host DCs for antigen presentation to induce optimal T cell immunity [5,6,7], and tumors often impair the function of host DCs to suppress anti-tumor immunity [8,9,10,11,12], DC-mediated immunosuppression presents a significant obstacle to successful DC vaccine development. Despite largely disappointing DC vaccine clinical trials, however, recent studies have shown that cDC1s play a critical role in cross-presenting tumor antigens to generate anti-tumor CD8 T cell immunity and in determining the efficacy of cancer immunotherapies including immune checkpoint blockade (ICB) [13,14,15]. Together with promising clinical results of DC vaccines with neoantigens [16], it is crucial to improve DC-based cancer vaccines to overcome DC-mediated immunosuppression, either by refining DC vaccines or combining DC vaccines with other therapies.

β-catenin, a major component of the Wnt signaling pathway, has emerged as a key factor in regulating DC differentiation and function [17,18,19]. We and others have shown that tumors, including melanoma, induce the upregulation/activation of β-catenin in DCs, and β-catenin promotes the tolerogenic function of DCs to suppress anti-tumor immunity [20,21,22,23,24]. However, the mechanisms by which β-catenin promotes DC-mediated immunosuppression remain incompletely understood. Besides β-catenin, inhibitory immune checkpoint molecules such as PD-L1 and Tim-3 are key players in promoting the tolerogenic function of DCs, and their expression on DCs plays a critical role in determining the efficacy of anti-PD-L1 and anti-Tim-3 immunotherapies, respectively [25,26,27]. Interestingly, β-catenin upregulates immune checkpoint molecules in tumor cells [28,29], raising the interesting scenario that β-catenin and checkpoint molecules might work collaboratively to modulate DC function. However, whether β-catenin in DCs regulates these inhibitory immune checkpoint molecules to suppress T cell responses has not been well studied [30]. 

In this report, we investigated whether β-catenin regulates inhibitory immune checkpoint molecules in DCs to exert its function. Based on our surprising finding that β-catenin upregulates Tim-3 in cDC1s, we aimed to determine whether β-catenin in DCs inhibits anti-tumor CD8 T cell responses through Tim-3 and whether combining Tim-3 blockade ICB with DC vaccines improves the anti-tumor efficacy of DC vaccines. 

## 3. Methods

### 3.1. Mice and Treatment

CD11c-β-catenin^active^ (CD11c-Cre^+^β-catenin^Exon3/Exon3^) mice were generated and maintained as previously described [20]. C57BL/6 mice and CD8 TCR transgenic Thy1.1^+^ Pmel-1 (gp100-specific CD8 T cells) mice were purchased from Charles River and Jackson Laboratory, respectively. Primary and recall responses were examined as previously described [20]. We cloned the melanoma tumor-associated antigen (TAA) human gp100 (hgp100) into anti-DEC-205-OVA constructs to generate anti-DEC-205-hgp100. For vaccinations, mice were injected intravenously (i.v.) with 15–40 μg of anti-DEC-205-hgp100, with CpG (100 μg) as an adjuvant. Anti-Tim-3 (300 μg/per mouse) was injected intraperitoneally in PBS on days −1, 1, and 3 after immunization. 

### 3.2. Antibodies and Reagents

A naive CD8 T cell isolation kit and Anti-Thy1.1 magnetic microbeads were purchased from Miltenyi Biotec (Auburn, CA, USA). Antibodies to Thy1.1 (clone OX-7, PerCP-Cy5.5), TCR Vα_2_ (clone B20.1, FITC), Vβ_5.1/5.2_ (clone MR9.4, PE), CD8α (clone 53.6.7, APC, PE-Cy7, BV510, and BV785), Siglec-H (clone 551, FITC, and PerCP-Cy5.5), Bst-2 (clone 927, PE, and Alexa Fluor 700), CD11b (clone M1/70, PE, and BV711), CD62L (clone MEL-14, FITC), CD44 (clone IM7, APC, and Alexa Fluor 700), CD11c (clone N418, APC, BV421, BV510, and APC-Cy7), MHC class II I-A^b^ (clone AF6-120.1, PerCP-Cy5.5), Tim-3 (clone RMT3.23, PE, APC, BV605, and PE-Dazzle 594), PD-L1 (clone 10F.9G2, PE, APC, PE-Cy7, BV510, APC-Cy7, and PE-Dazzle 594), PD-L2 (clone TY25, PE, PE-Cy7, BV421, and PE-Dazzle 594), Lag-3 (clone C9B7W, PE, PE-Dazzle 594, and BV785), PD-1 (clone 29F.1A12, PE, APC, PE-Cy7, BV510, and APC-Cy7), CCR7 (clone 4B12, PE, APC, PE-Cy7, and PE-Dazzle 594), IFN-γ (clone XMG1.2, APC, and Alexa Fluor 700), TNF-α (clone MP6-XT22, PE, BV421, and BV605), IL-2 (clone JES6-5H4, APC, BV510, and PE-Dazzle 594) and Granzyme B (clone QA16A02, PE, and Alexa Fluor 700), were purchased from Biolegend Inc. (San Diego, CA, USA). Staining for surface and intracellular antigen expression was performed as previously described [17]. In brief, cells from the spleen or pooled draining LNs were stimulated for 5 hours with hgp100_25–33_ peptide (4 μg/mL, AnaSpec, Fremont, CA, USA) in the presence of Brefeldin A (BFA, 5 μg/mL, Biolegend), stained for surface and intracellular (e.g. IL-2, TNFα and IFN-γ) antigens, and evaluated by flow cytometry. Where indicated, Pmel-1 CD8 T cells were labeled with 5-(6)-carboxyfluorescein diacetate succinimidyl diester (CFSE) and checked by flow cytometry before transfer. We used a Celesta™ (BD Biosciences, Franklin Lakes, NJ, USA) or NovoCyte Quanteon (Agilent Technology, Santa Clara, CA, USA) with subsequent analysis of data in FlowJo^®^ (Tree Star, Ashland, OR, USA).

### 3.3. In Vivo Cross-Priming Assays

In vivo cross-priming assays were performed as described previously [20]. Briefly, 0.5–1 × 10^6^ CFSE-labeled naïve Pmel-1 Thy1.1^+^ CD8^+^ T cells were injected intravenously by the tail vein in 200 μL PBS −1, 0, or 1 day after immunization with anti-DEC-205-hgp100, and 5 days later, spleen and LN cells were subjected to staining and flow cytometry as described above.

### 3.4. Single-Cell RNA Sequencing (scRNAseq) of Primed Pmel-1 CD8 T Cells

For isolation of adoptively transferred Pmel-1 Thy1.1^+^ CD8^+^ T cells, spleen cells from immunized mice were first enriched with anti-Thy1.1 magnetic microbeads and sorted. Then, a 10X Genomics Chromium Single-Cell 3′ Reagent Kit (v2 Chemistry) and Chromium Single-Cell Controller were used to generate scRNA-seq libraries, as detailed in our previously published study [31]. Briefly, cells sorted by FACS were loaded into each reaction for gel bead-in-emulsion (GEM) generation and cell barcoding. A Veriti™ 96-Well Fast Thermal Cycler (Applied Biosystems, Waltham, MA, USA) was used to reverse transcribe the GEM (GEM-RT) at 53 °C for 45 min, 85 °C for 5 min, followed by a 4 °C hold. Following GEM-RT cleanup using Dynabeads MyOne Silane (Thermo Fisher Scientific, Waltham, MA, USA), cDNA amplification was carried out with the same Thermal Cycler using the following program: 98 °C for 3 min, 98 °C for 15 s, 67 °C for 20 s, and 72 °C for 1 min, followed by a 12-cycle repeat at 72 °C for 1 min then held at 4 °C. Amplified cDNA was cleaned up with a SPRIselect Reagent Kit (Beckman Coulter, Brea, CA, USA), which was followed by a library construction procedure, including fragmentation, end repair, adapter ligation, and library amplification. The quality of the libraries was assessed using an Agilent 2100 Bioanalyzer (Agilent, Santa Clara, CA, USA). Subsequently, the libraries underwent sequencing on an Illumina HiSeq4000 platform using a paired-end flow cell as follows: Read 1, 26 cycles; i7 index, 8 cycles; Read 2, 98 cycles.

### 3.5. Data Analysis for scRNAseq

scRNAseq data were conducted as previously described [32]. The following steps were undertaken for the analysis of sequenced reads from scRNAseq libraries: demultiplexing, alignment to the mm10 mouse reference, barcode processing, and Unique Molecular Identifier (UMI) counting using the 10X Genomics Cell Ranger (v2.0.1) pipeline [33]. A total of 6752 cells from the CD11c-β-catenin^active^ mice and 10,254 cells from the WT mice were analyzed for a total of 16,685 genes. The datasets were then processed using the R Seurat package [34,35], with Principle Component Analysis (PCA) being employed for combined samples (cells from CD11c-β-catenin^active^ mice and WT) analysis. The quality control metrics employed were as follows. Two strategies were employed to identify potential doublets as follows: cells expressing both X and Y chromosome-linked genes (Kdm5d, Eif2s3y, Gm29650, Uty, and Ddx3y) were excluded, as were cells expressing unusually high numbers of genes (>4000); low-quality cells with a low number of genes (<300) and/or having high mitochondrial genetic content (>5%) were excluded. Additionally, uninteresting sources of variation were removed, including ribosomal structural proteins (as identified by gene ontology term GO:0003735 and the Ribosomal Protein Gene (RPG) database [36]), non-coding rRNAs, Hbb, and genes not expressed in more than 3 cells. 

For gene expression normalization, the Seurat package’s “LogNormalize” method was employed, normalizing each cell’s gene expression measurements by total expression, multiplied by a factor of 10,000, followed by log transformation. Highly variable genes were identified in each dataset, with the top 1000 genes intersecting between datasets being utilized for clustering and subsequent analyses. The number of principal components (PCs) used for cell clustering was determined using the Jackstraw method, while the number of canonical correlation components (CCs) was determined by manual inspection of scree plots. Following the identification of PC and CC numbers for downstream analyses (first 30 PCs), a graph-based clustering approach in Seurat was used to iteratively cluster cells based on component similarities. The resulting clusters were visualized using the uniform manifold approximation and projection (UMAP) method. To assess the potential effects of cell cycle heterogeneity in the clustering, cell cycle phase scores were calculated based on canonical markers and regressed from the data [37]. The *FindAllMarkers* function in Seurat was then employed to identify differentially expressed genes (DEGs) between clusters, with a Bonferroni adjustment of *p*-value < 0.05 set as a statistical significance threshold. DEGs of CD11c-β-catenin^active^ mice and WT mice were identified with a fold change (FC) greater than 1.2, with an adjusted *p*-value less than 0.05. DAVID Tool [38] was employed for Gene Ontology (GO) analysis.

### 3.6. Tumor Cell Lines and Treatment of Tumor-Bearing Mice

B16F10 melanoma cells were inoculated by subcutaneous (s.c.) injection. Once the tumors became palpable, they were monitored every other day, and tumor sizes were calculated as (0.5 × short length × long length × long length mm^3^). For treatments, increased tumor volume was calculated as tumor volume on the day of measurement subtracting the tumor volume at the time of immunization. Mice were euthanized if signs of illness were observed or when any one dimension of the tumor exceeded 20 mm.

### 3.7. Statistical Analysis

Excel or GraphPad Prism 9 was used to evaluate data statistical significance with the two-tailed unpaired two-sample student’s *t*-test for two groups or ANOVA with post hoc tests for 3 or more groups. The mean and standard error of the mean (SEM) were presented. Data were considered significant if *p*-values were less than 0.05.

## 4. Results 

### 4.1. β-Catenin Upregulates the Expression of the Inhibitory Molecule Tim-3 in DCs

We have shown previously that activation of β-catenin in DCs negatively regulates both CD4 and CD8 T cell immune responses [17,20], although the mechanisms remain incompletely understood. Since β-catenin upregulates immune checkpoint molecules in tumor cells [28,29], we asked whether the expression of inhibitory immune checkpoint molecules was affected by β-catenin^active^ DCs, leading to impaired DC function. Spleen and lymph node (LN) cells from WT and CD11C-β-catenin^active^ mice were subjected to flow cytometry to determine the expression of the inhibitory molecules PD-L1, PD-L2, Tim-3, and Lag3 on DCs. As cDC1s play a critical role in cross-presenting tumor antigens to generate anti-tumor CD8 T cell immunity and we could successfully separate pDCs, cDC1s, and cDC2s in splenocytes (Appendix A), we focused our analysis on splenic cDC1s. Activation of β-catenin in DCs led to an increase in the CD8α^+^ cDC1 population within all the cDCs (Figure 1A), suggesting that the impaired induction of CD8 T cell responses in CD11C-β-catenin^active^ mice is not due to a reduced number of cDC1s. The increased cDC1 frequency in CD11C-β-catenin^active^ mice is consistent with previous reports [20,39]. We next examined the expression of the inhibitory immune checkpoint molecules on cDC1s. Notably, Tim-3 expression was significantly upregulated in β-catenin^active^ cDC1s compared with WT cDC1s (Figure 1B,C), while expression of other inhibitory molecules was not elevated (PD-L1 and Lag3) or only slightly increased (PD-L2) in β-catenin^active^ DCs (Appendix A). Taken together, these data indicate that β-catenin upregulates Tim-3 in cDC1s.

### 4.2. β-Catenin in DCs Suppresses Tumor Antigen-Specific CD8 T Cell Responses

To examine tumor antigen-specific CD8 T cell responses, we constructed an anti-DEC-205 antibody expressing the melanoma antigen human gp100 (anti-DEC-205-hgp100) and asked whether β-catenin in DCs regulated tumor antigen-specific CD8 T cell responses. WT and CD11c-β-catenin^active^ mice were adoptively transferred with gp100-specific Pmel-1 Thy1.1^+^ CD8 T cells and immunized with anti-DEC-205-hgp100 plus adjuvant CpG one day later. The percentages of Thy1.1^+^ Pmel-1 out of total CD8 T cells, as well as differentiated IFN-γ^+^Thy1.1^+^ Pmel-1 effectors out of total Pmel-1 CD8 T cells, were significantly lower in CD11C-β-catenin^active^ mice compared with WT mice for both spleen and LN cells (Figure 2A,B), suggesting that activation of β-catenin in DCs diminished gp100-specific CD8 T cell proliferation and differentiation into effectors. Furthermore, IFN-γ^+^IL-2^+^ polyfunctional Pmel-1 effectors producing IL-2 in addition to IFN-γ, which are critical for CD8 T cell memory responses [40,41], were significantly reduced in CD11C-β-catenin^active^ mice compared with WT mice (Appendix A). Interestingly, the expression of inhibitory immune checkpoint molecules including Tim-3, CTLA-4, PD-1, and Lag-3 was not significantly increased in gp100-specific CD8 T cells primed in CD11C-β-catenin^active^ mice compared with WT mice (Appendix A). In fact, gp100-specific Pmel-1 CD8 T cells primed in CD11C-β-catenin^active^ mice generally exhibited lower expression of these inhibitory molecules, with some reaching statistical significance (Appendix A), suggesting that these inhibitory molecules are unlikely to account for impaired effector differentiation of gp100-specific Pmel-1 CD8 T cells.

We next asked whether β-catenin activation in DCs also negatively regulated gp100-specific CD8 T cell memory responses. WT and CD11c-β-catenin^active^ mice were immunized as above and then challenged on day 21 with the human gp100 epitope (gp100_25–33_) to assess memory CD8 T cell responses. Consistent with our data on the cross-priming of Pmel-1 cells, total Thy1.1^+^ Pmel-1 and IFN-γ^+^ Thy1.1^+^ Pmel-1 effector cells in spleen and draining LN cells were greatly reduced in CD11C-β-catenin^active^ mice, suggesting that β-catenin activation in DCs results in dampened CD8 recall responses (Figure 2C,D). Taken together, these data demonstrate that β-catenin in DCs negatively regulates primary and memory tumor antigen-specific CD8 T cell responses.

### 4.3. Single-Cell RNA Sequencing of Pmel-1 CD8 T Cells Primed in WT and CD11c-β-Catenin^active^ Mice

To better understand how β-catenin in DCs affects the differentiation of gp100-specific Pmel-1 CD8 T cells into effector and memory CD8 T cells, we performed single-cell RNA sequencing (scRNA-seq) of sorted Thy1.1^+^ Pmel-1 CD8 T cells from WT and CD11c-β-catenin^active^ mice on day 10 after immunization with anti-DEC-205-hgp100 plus CpG. Pmel-1 CD8 T cells were clustered based on gene expression using an unsupervised inference analysis. There were a total of nine clusters that were identified and visualized by the uniform manifold approximation and projection for dimension reduction (UMAP) algorithm [42] (Figure 3A). Our clustering data analysis showed that primed Pmel-1 CD8 T cells fall into two major populations (cluster 0 and cluster 1) with similar frequencies in both WT and CD11c-β-catenin^active^ mice (Figure 3A,B). However, Pmel-1 cells from WT and CD11c-β-catenin^active^ mice exhibited dramatic differences in the percentages of Pmel-1 CD8 T cells in clusters 2 and 4 and a small difference in cluster 3. Specifically, CD11c-β-catenin^active^ mice showed a substantially lower proportion of Pmel-1 cells in clusters 2 and 4 compared with WT mice and slightly more cells in cluster 3 compared with WT mice (Figure 3A,B). Further analysis of the top differentially expressed genes (DEGs) revealed a clear separation of the nine heterogenous Pmel-1 cell populations following immunization (Figure 3C). These clusters exhibit distinct gene expression profiles, for example, cluster 8 is distinguished by its expression of complement-related genes (*Cd5l*, *C1qa*, *C1qc*), whereas cluster 7 represents a rare ‘B-T’ cluster characterized by elevated expression of B cell markers such as *Cd19*, *Ms4a1(CD20*), and *Cd79a* (Figure 3C), as reported in previous studies [43,44,45]. Cluster 5 consists of CD8 T cells with heightened expression of IFN-responsive genes, such as Ifit1, Ifit3, and Igs15 [46]. Clusters 1, 2, and 3 stand out from other clusters because of their increased expression of a distinct gene set, as shown in Figure 3C. While cluster 4 Pmel-1 cells exhibit shared expression of DEGs observed in clusters 1, 2, and 3, they are also distinguished from other clusters by their unique expression of cell cycle genes (*Hist1h3c*, *Hmmr*, *Esco2*, *Kif4*, *Ska1*, *Ccna2*, *Pclaf*, and *Cmtm7*) (Figure 3C), suggesting that they likely represent proliferating Pmel-1 CD8 T cells.

Considering that the fraction of Pmel-1 cell clusters only differed between CD11c-β-catenin inactive mice and WT mice in clusters 2, 3, and 4, we focused on these clusters and visualized their top three most DEGs as violin plots. As shown in Figure 3D, clusters 3 and 4 exhibited high expression of effector markers (*Ccl4*, *Ccl5*, and *Infg*), while cluster 2 displayed relatively high expression of activation markers such as *S100a6*, *Slamf6*, and *Ctla2a*. Intriguingly, in addition to high levels of both effector and activation makers, cluster 4 also exhibited high expression of cell cycle genes, such as *Hmmr*, *Mik67*, and *Pclaf* (Figure 3D), indicating that these Pmel-1 cells represent highly activated effector T cells. A significantly reduced induction of these proliferating cluster 4 Pmel-1 cells in CD11c-β-catenin^active^ mice (Figure 3A,B) likely leads to impaired memory Pmel-1 cell differentiation, resulting in reduced memory responses (Figure 2C,D). Further analysis revealed a particularly high effector score located in both cluster 3 and cluster 4, which was significantly increased in Pmel-1 cells in WT compared with CD11c-β-catenin^active^ mice (Figure 3E). In line with this, the expression of selected effector genes (*Gzmb*, *Ifng*, and *Prf1*) further confirmed that Pmel-1 cells in WT mice were elevated in WT Pmel-1 cells, especially in clusters 3 and 4. 

Consistent with increased effector T cell priming, Pmel-1 T cells in WT mice exhibited an overall higher memory score compared with Pmel-1 T cells in CD11c-β-catenin^active^ mice, although the difference was not as striking as for the effector score (Figure 3E). We thus confirmed that cluster 3 and cluster 4 expressed transcriptional programs of both effector and memory CD8 T cells (Figure 3E), corroborating previous reports on vaccine-primed CD8 T cells [47,48]. Consistently, the expression of individual memory markers (*Sell*, *Tcf7*, and *Id3*) was higher in Pmel-1 cells primed in WT mice, although more evenly distributed across the different clusters (Figure 3E). 

We next carried out Gene Ontology (GO) enrichment analysis on Pmel-1 cells primed in WT and CD11c-β-catenin^active^ mice, specifically focusing on pathways that were significantly downregulated in Pmel-1 cells primed in CD11c-β-catenin^active^ mice. As expected, biological processes related to T cell-mediated cytotoxicity, as well as molecular functions associated with TCR/CD8 receptor binding, were significantly reduced in Pmel-1 cells in CD11c-β-catenin^active^ mice (Figure 3F). Additionally, processes linked to mitochondrial energy generation were significantly reduced (Figure 3F), which is consistent with diminished effector T cell differentiation, as shown in Figure 3E. Notably, processes related to cell division/cycle were also significantly decreased (Figure 3F), strongly suggesting that Pmel-1 cells in CD11c-β-catenin^active^ mice are less proliferative. This is in agreement with the almost complete loss of proliferative cluster 4 among Pmel-1 cells in CD11c-β-catenin^active^ mice (Figure 3A,B). Supporting this notion, proteasome (core) complexes, which play a central role in the regulation of proteins that control cell-cycle progression, were identified as the most downregulated cellular components in CD11c-β-catenin^active^ mice (Figure 3F). Thus, our GO enrichment analysis similarly identified transcriptionally downregulated GO terms associated with effector function and proliferation of primed Pmel-1 CD8 T cells through interaction with β-catenin^active^ DCs.

### 4.4. Anti-Tim-3 Treatment Reverses β-Catenin-Mediated Suppression of CD8 T Cell Responses

After showing that cDC1s with active β-catenin exhibit augmented expression of Tim-3, we examined whether blocking Tim-3 by an anti-Tim-3 antibody could restore cross-priming in CD11c-β-catenin^active^ mice. WT and CD11c-β-catenin^active^ mice were adoptively transferred with gp100-specific Pmel-1 CD8 T cells and immunized with anti-DEC-205-hgp100, as described above. Half of the immunized CD11c-β-catenin^active^ mice were also treated with anti-Tim-3. As expected, CD11c-β-catenin^active^ mice exhibited impaired cross-priming, showing reduced total Thy1.1^+^ Pmel-1 cells and IFN-γ^+^Thy1.1^+^ Pmel-1 effectors (Figure 4A,B). However, treating CD11C-β-catenin^active^ mice with the anti-Tim-3 antibody restored cross-priming, resulting in percentages of total Thy1.1^+^ Pmel-1 cells and IFN-γ^+^Thy1.1^+^ effectors comparable to the percentages in WT mice (Figure 4A,B). Interestingly, polyfunctional Pmel-1 effectors that produced IL-2 in addition to IFN-γ were completely restored in CD11C-β-catenin^active^ mice by anti-Tim-3 treatment compared with WT mice (Figure 4C). Moreover, treatment of CD11c-β-catenin^active^ mice with the anti-Tim-3 antibody did not lead to reduced expression of Tim-3 or other inhibitory immune molecules on Pmel-1 cells (Appendix A). Together with the fact that the expression of these inhibitory immune checkpoint molecules was not significantly increased in gp100-specific CD8 T cells primed in CD11C-β-catenin^active^ mice (Appendix A), it seems unlikely that inhibitory immune checkpoint molecules on Pmel-1 CD8 T cells account for the effects of anti-Tim-3 on cross-priming. Taken together, our data suggest that β-catenin regulates DC vaccine-induced cross-priming through Tim-3, and Tim-3 blockade reverses β-catenin-mediated inhibition of cross-priming. 

We next examined memory responses upon treatment with the anti-Tim-3 antibody. WT and CD11c-β-catenin^active^ mice were adoptively transferred with gp100-specific Pmel-1 CD8 T cells and immunized with anti-DEC-205-hgp100 alone or together with anti-Tim-3 as above. Immunized mice were then challenged with the human gp100 epitope (gp100_25–33_) on day 21 to assess memory CD8 T cell responses. As expected (see Figure 2C,D), total Thy1.1^+^ Pmel-1 and IFN-γ^+^ Thy1.1^+^ Pmel-1 effector cells were significantly reduced in CD11C-β-catenin^active^ mice (Figure 5A,B). Notably, treating CD11C-β-catenin^active^ mice with the anti-Tim-3 antibody restored recalled memory response for gp100-specific Pmel-1 CD8 T cells, as the percentages of total Thy1.1^+^ Pmel-1 cells and IFN-γ^+^Thy1.1^+^ effectors were comparable to the percentages in WT mice (Figure 5A,B). Taken together, our data indicate that blocking Tim-3 reverses β-catenin-mediated suppression of primary and memory tumor antigen-specific CD8 T cell responses upon DC vaccination. 

### 4.5. Combination of Tim-3 Blockade and DC Vaccination Led to Improved DC Vaccine Efficacy

So far, our experiments have established that β-catenin in DCs plays a negative role in regulating tumor antigen-specific CD8 T cell responses and that β-catenin-mediated suppression of cross-priming could be reversed by Tim-3 blockade. Therefore, we asked whether the combination of Tim-3 blockade and DC vaccines will lead to improved anti-tumor efficacy. B16F10-bearing WT mice were either left untreated or immunized with anti-DEC-205-hgp100 following Pmel-1 CD8 T cell transfer, with half of the vaccinated mice also treated with anti-Tim-3. As expected, B16F10-bearing WT mice vaccinated with anti-DEC-205-hgp100 exhibited significantly slower tumor growth compared with untreated B16F10-bearing WT mice (Figure 6A). Notably, vaccinated B16OVA-bearing mice also treated with the anti-Tim-3 antibody exhibited much slower tumor growth compared with mice with anti-DEC-205-hgp100 vaccination alone (Figure 6A), resulting in substantially smaller tumor sizes compared with unvaccinated mice or mice with only anti-DEC-205-hgp100 vaccination (Figure 6B,C). Thus, blocking Tim-3 during DC-targeted vaccination improves DC vaccine efficacy. 

## 5. Discussion

Here, we identified the β-catenin/Tim-3 axis in cDC1s as a new mechanism for β-catenin in DCs to inhibit tumor antigen-specific CD8 T cell immunity. In this report, we demonstrated that activation of β-catenin upregulates Tim-3 in cDC1s, the DC subset specialized in cross-presenting tumor antigens to generate anti-tumor immunity and in determining the efficacy of cancer immunotherapies. Employing a cDC1-targeted vaccine model using the melanoma antigen human gp100 (hgp100), we further demonstrated that treatment with anti-Tim-3 restored cross-priming and memory responses of gp100-specific CD8 T cells in CD11c-β-catenin^active^ mice. Notably, combination therapy of the DC-targeted vaccine with anti-Tim-3 antibody treatment led to improved anti-tumor efficacy, thus supporting the β-catenin/Tim-3 axis in cDC1s as a new target for therapeutic intervention. 

Our previous studies have shown that activation of β-catenin in DCs either genetically or by tumors suppresses cDC1-targeted vaccine-induced anti-tumor CD8 T cell immunity using ovalbumin as a model antigen [20]. In this report, we constructed an anti-DEC-205 antibody expressing the human melanoma antigen hgp100 and examined CD8 T cell responses against a genuine tumor antigen. hgp100 is a homologue of the mouse self/tumor antigen mgp100 and contains the altered peptide ligand hgp100_25–33_, which can elicit gp100-specific CD8 T cell immunity in the B16 melanoma model [49,50,51]. Extending our previous findings, we showed that β-catenin in DCs similarly suppressed cross-priming and memory responses of gp100-specific Pmel-1 CD8 T cell responses upon cDC1-targeted vaccination with anti-DEC-205-hgp100 (Figure 2). We further analyzed primed gp100-specific Pmel-1 CD8 T cells from vaccinated WT and CD11c-β-catenin^active^ mice using scRNA-seq. Our scRNA-seq transcriptional data identified nine distinct clusters with varied expression of effector and memory/stem-like cell markers (Figure 3A,E), confirming the heterogeneity in tumor antigen-specific CD8 T cell responses after cDC1-targeted vaccination. Of note, all these clusters exhibited high expression of the memory/stem-like cell marker TCF1 (*Tcf7*), consistent with previous studies with vaccines with peptides and peptide-pulsed DCs [47,52]. Among these clusters, Pmel-1 cells in clusters 3 and 4 (and to a lesser extent, cluster 2) maintained the expression of both effector and memory markers (Figure 3), resembling populations that have been reported among vaccine-primed antigen-specific CD8 T cells [47,48]. Strikingly, while cluster 4 Pmel-1 cells display high levels of effector and activation makers, they are the only cells that express cell cycle genes (Figure 3C,D), indicating that this particular subset of Pmel-1 cells represents proliferative effector cells that likely give rise to memory cells. Indeed, Pmel-1 cells in CD11c-β-catenin^active^ mice contained a substantially reduced cluster 4 compared with that of WT mice (Figure 3B), consistent with their impaired memory responses (Figure 2C,D). Supporting this notion, GO enrichment analysis of our scRNA-seq data identified both cell cycle-related GO terms and CD8 T cell function-related GO terms that were transcriptionally downregulated by interaction with β-catenin^active^ DCs (Figure 3F). Our scRNA-seq findings thus indicate that β-catenin in DCs negatively regulated transcription programs governing the effector and memory differentiation of antigen-specific CD8 T cells, corroborating the observed suppressed cross-priming and memory responses in CD11c-β-catenin^active^ mice. Taken together, our data support a model in which β-catenin in DCs regulates transcription programs in primed antigen-specific CD8 T cells to impair their effector functions and the generation of memory CD8 T cells, resulting in suppressed CD8 T cell immunity. 

To date, studies of the role of β-catenin in DCs by us and others have supported its role in promoting a tolerogenic function of DCs through a number of mechanisms [17,18,20,21,22,23,24,53,54]. For example, we reported that blocking β-catenin or its downstream IL-10 during the T cell priming phase led to improved anti-tumor CD8 T cell responses by DC vaccines [22], and recent studies found that blocking β-catenin synergized with anti-PD-1 immunotherapy [55,56]. On the other hand, we previously reported that β-catenin in DCs also played a positive role in the maintenance of DC vaccine-induced CD8 T cell memory responses [22], suggesting that β-catenin might not be an ideal therapeutic target to boost anti-tumor CD8 T cell immunity. Intriguingly, β-catenin has been shown to upregulate checkpoint molecules on tumor cells [28,29], but whether β-catenin similarly controls checkpoint molecules on DCs to exert its immunoregulatory function remains unclear. We were surprised to observe that β-catenin^active^ cDC1s (cDC1s with active β-catenin, from CD11c-β-catenin^active^ mice) exhibited elevated expression of Tim-3 (Figure 1) in contrast to unchanged or reduced expression of other inhibitory molecules including PD-L1 and Lag3 (Appendix A). The anti-Tim-3 antibody has been tested clinically as a new addition to immune checkpoint blockade immunotherapies [57,58,59], although the mechanism underlying how Tim-3 blockade achieves its anti-tumor effects is not well understood [60]. Of note, we and others have shown that multiple tumors induce up-regulation/activation of β-catenin in DCs, including tumor-infiltrated DCs to restrain anti-tumor CD8 T cell responses [20,21,23,24], raising the intriguing scenario that tumors might exploit the β-catenin/Tim-3 pathway in DCs to suppress anti-tumor CD8 T cell immunity.

Tim-3 was originally identified as a cell surface marker of IFN-γ-producing CD4 T helper cells and cytotoxic CD8 T cells [61], although recent studies have shown that Tim-3 is also constitutively expressed on DCs [62]. Indeed, cDC1s, the DCs that are critical in cross-presenting tumor antigens to generate anti-tumor CD8 T cell immunity, express the highest level of Tim-3 among DCs [62,63,64]. In addition, tumor-associated DCs (TADCs) in multiple tumors express high levels of Tim-3 [62,63,64], and the effects of anti-Tim-3 antibody treatment in improving the anti-tumor efficacy of chemotherapy and DC vaccines are dependent on DCs [62,65]. In line with these studies, a recent report has further demonstrated that Tim-3 on DCs instead of T cells (both CD4 and CD8 T cells) is essential for mediating the beneficial effects of anti-tumor immunity by Tim-3 blockade [27]. Given that β-catenin activation led to the upregulation of Tim-3 on cDC1s (Figure 1), we first asked whether blocking Tim-3 would reverse the β-catenin-mediated suppression of antigen-specific CD8 T cell responses. Indeed, treatment with anti-Tim-3 completely restored the cross-priming and memory responses of gp100-specific CD8 T cells in CD11c-β-catenin^active^ mice (Figure 4 and Figure 5), suggesting that Tim-3 could serve as a therapeutic target to improve DC vaccines. On the other hand, Tim-3 expression (as well as other inhibitory immune checkpoint molecules such as CTLA-4, PD-1, and Lag3) on primed Pmel-1 CD8 T cells was not enhanced in CD11c-β-catenin^active^ mice (Appendix A). Together with the fact that anti-Tim-3 treatment did not reduce the expression of these molecules on primed Pmel-1 CD8 T cells in CD11c-β-catenin^active^ mice (Appendix A), our data suggest that Tim-3 or the other inhibitory immune checkpoint molecules on Pmel-1 CD8 T cells are unlikely to be a major contributor to the positive effects of anti-Tim-3 treatment on cross-priming. Supporting this notion, immune checkpoint molecules including Tim-3, PD-1, and Lag3 were only minimally detected on primed Pmel-1 cells by our scRNA-seq data analysis. Taken together, our findings indicate that Tim-3 on DCs (and not T cells) plays a critical role in mediating the effects of anti-Tim-3 antibody treatment on DC vaccine-induced cross-priming. These results align with previous studies highlighting the pivotal involvement of Tim-3 on DCs in driving the anti-tumor effects resulting from Tim-3 blockade [27,62,65]. 

It should be noted that our findings with Tim-3 do not exclude the possible involvement of other inhibitory molecules in the DC β-catenin-mediated regulation of CD8 T cell responses. Indeed, we showed that β-catenin^active^ cDC1s also exhibited slightly but significantly higher PD-L2 expression and significantly lower PD-L1 expression (Appendix A). Further studies are warranted to investigate the roles of these molecules in the β-catenin-mediated inhibition of CD8 T cell immunity. Our studies also did not address how anti-Tim-3 treatment restores DC vaccine-induced cross-priming and memory CD8 T cell responses in CD11c-β-catenin^active^ mice. However, given that Tim-3 on DCs instead of T cells mediates the effects of Tim-3 blockade [27], together with our findings that anti-Tim-3 treatment did not affect Tim-3 expression on primed CD8 T cells (Appendix A), it is likely that anti-Tim-3 exerts its function by targeting Tim-3 on DCs. As previous studies have shown the specificity of β-catenin expression in DCs instead of other immune cells in CD11c-β-catenin^active^ mice [18,39], our findings suggest a potential β-catenin-Tim-3 axis in DCs that regulates DC-vaccine-induced CD8 T cell responses. Interestingly, previous studies have shown that anti-Tim-3 treatment regulates the function of cDC1s by augmenting their expression of Cxcl9, leading to enhanced CD8 T cell responses [62]. We previously reported that β-catenin^active^ DCs produced much higher IL-10 upon CpG treatment, which in turn inhibits DC vaccine-induced CD8 T cell responses [22]. It will be interesting to examine whether anti-Tim-3 treatment reverses β-catenin-mediated inhibition of CD8 T cell responses by regulating these cytokine pathways in DCs. One limitation of our study is that we only examined the effects of anti-Tim-3 treatment in the setting of DC vaccination with anti-DEC-205 plus CpG. Given that infections and adjuvants affect the effects of β-catenin on CD8 T cell responses [20,39], it is important to expand our study to investigate how anti-Tim-3 treatment regulates the β-catenin-mediated regulation of CD8 T cell responses under different settings. 

We further investigated the anti-tumor effects of combining DC-targeted vaccines with anti-Tim-3 treatment. Anti-Tim-3 treatment significantly improved the anti-tumor efficacy of DC vaccines, leading to a much-reduced tumor growth and tumor mass (Figure 6). As anti-Tim-3 treatment alone has limited or no anti-tumor efficacy in the B16 melanoma model [66,67,68,69], these data suggest that Tim-3 blockade synergizes with DC vaccines to improve their anti-tumor efficacy. In conclusion, we identified the β-catenin/Tim-3 axis as a novel mechanism that inhibits DC-mediated CD8 T cell responses, thereby supporting Tim-3 as a new target for therapeutic intervention to improve the anti-tumor efficacy of DC vaccines. 

## 6. Conclusions

In this report, we have found Tim-3 is upregulated by β-catenin in cDC1s. Using a cDC1-targeted vaccine model, we have demonstrated that mice with active β-catenin in their DCs (CD11c-β-catenin^active^ mice) exhibited impaired cross-priming and memory responses of tumor antigen-specific CD8 T cells. Anti-Tim-3 antibody treatment restored cross-priming in CD11C-β-catenin^active^ mice. Furthermore, treating B16F10-bearing mice with a DC vaccine using anti-DEC-205-hgp100 in combination with the anti-Tim-3 antibody led to significantly reduced tumor growth compared with treatment with the DC vaccine alone, thus identifying the β-catenin-Tim-3 pathway as a new target for therapeutic intervention. Taken together, we have identified a β-catenin/Tim-3 axis in DCs that negatively regulates anti-tumor CD8 T cell immunity and that combination immunotherapy of a DC-targeted vaccine with anti-Tim-3 antibody leads to improved anti-tumor efficacy. 

## Figures and Tables

**Figure 1 vaccines-12-00460-f001:**
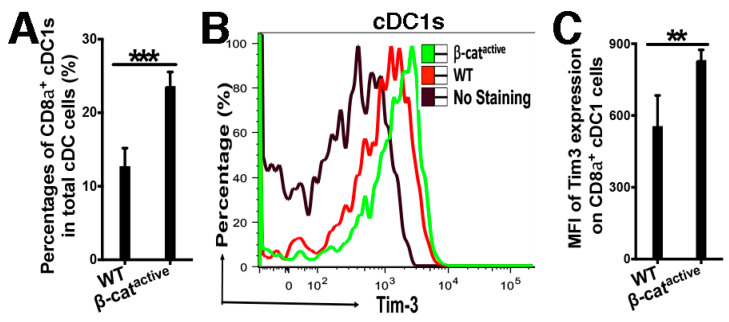
β-catenin upregulates Tim-3 in splenic cDC1s. (**A**) CD11C-β-catenin^active^ mice exhibit higher percentages of splenic cDC1s than WT mice (*n* = 4). Splenic cells were stained and analyzed by flow cytometry, and the percentages of CD8α^+^ cDC1s out of the total cDCs (95% intervals: 12.7 ± 2.5% vs. 23.6 ± 1.9%) are shown. (**B**,**C**) β-catenin^active^ cDC1s express significantly higher Tim-3 than WT cDC1s by flow cytometry. Histogram overlay of Tim-3 expression (**B**) and Mean Fluorescence Intensity (MFI) of Tim-3 expression (**C**) on gated cDC1s (95% intervals for MFI: 555 ± 128 vs. 832 ± 43) are shown. Student’s *t*-test, *** *p* < 0.001, ** *p* < 0.01. The data shown are representative of at least three experiments.

**Figure 2 vaccines-12-00460-f002:**
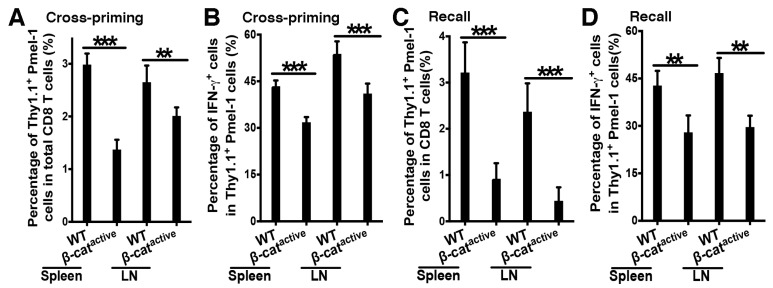
β-catenin in DCs negatively regulates DC vaccine-induced CD8 T cell responses against tumor antigens. WT and CD11c-β-catenin^active^ mice were immunized with anti-DEC-205-hgp100 plus CpG, and cross-priming of adoptively transferred Thy1.1^+^ Pmel-1 CD8 T cells was examined on day 5 following a 5-hour in vitro stimulation with hgp100_25–33_ with Brefeldin A (BFA) (**A**,**B**), recall responses were examined on day 26 after recall with hgp100_25–33_ peptide on day 21 (**C**,**D**). (**A**,**B**) CD11c-β-catenin^active^ mice exhibit impaired cross-priming of adoptively transferred hgp100-specific (Pmel-1) CD8 T cells upon DC-targeted vaccination compared with WT mice (*n* = 4–5). (**A**) The percentages of Thy1.1^+^ Pmel-1 cells in total CD8 T cells (95% intervals: spleen, 3.0 ± 0.2% vs. 1.4 ± 0.2%; LN, 2.7 ± 0.3% vs. 2 ± 0.2%) and (**B**) the percentages of IFN-γ^+^ effectors out of total Thy1.1^+^ Pmel-1 cells (95% intervals: spleen, 43.4 ± 1.8% vs. 31.8 ± 1.7%; LN, 53.8 ± 4% vs. 41 ± 3.2%) on day 5 following a 5-hour in vitro stimulation with hgp100_25–33_ with Brefeldin A (BFA) are shown. Student’s *t*-test, *** *p* < 0.001. (**C**,**D**) CD11c-β-catenin^active^ mice exhibit impaired recall responses compared with WT mice (*n* = 5). (**C**) The percentages of Thy1.1^+^ Pmel-1 cells in total CD8 T cells (95% intervals: spleen, 3.2 ± 0.7% vs. 0.9 ± 0.3%; LN, 2.4 ± 0.6% vs. 0.4 ± 0.3%) and (**D**) the percentages of IFN-γ^+^ effectors out of total Thy1.1^+^ Pmel-1 cells (95% intervals: spleen, 42.8 ± 4.7% vs. 28 ± 5.3%; LN, 46.7 ± 4.8% vs. 29.6 ± 3.6%) on day 5 after recall are shown. Student’s *t*-test, ** *p* < 0.01 and *** *p* < 0.001. The data shown are representative of two or more experiments.

**Figure 3 vaccines-12-00460-f003:**
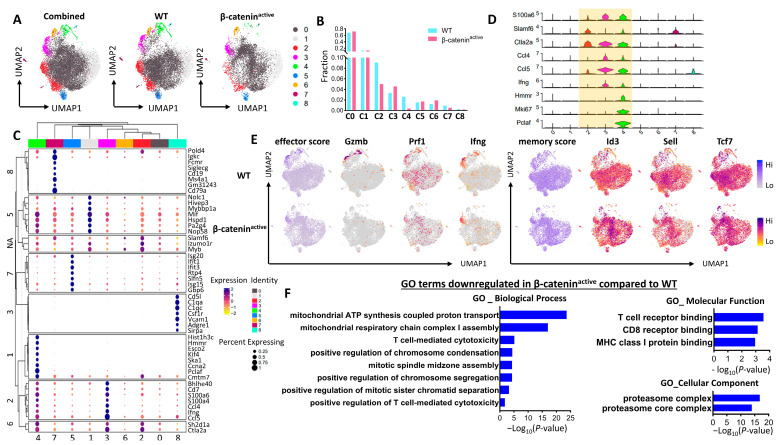
scRNA-seq of gp100-specific CD8 T cells identifies distinct populations and reveals differences in CD8 differentiation between CD8 T cells primed in WT and CD11c-β-catenin^active^ mice. WT and CD11c-β-catenin^active^ mice adoptively transferred Thy1.1^+^ Pmel-1 CD8 T cells were immunized with anti-DEC-205-hgp100 plus CpG. Spleen cells were harvested on day 10 after immunization, and FACS-sorted Pmel-1 cells were subjected toscRNA-seq as described. (**A**) Uniform Manifold Approximation and Projection (UMAP) dimensionality reduction mapping analysis of single-cell gene expression data of Pmel-1 cells isolated 10 days after vaccination with ant-DEC-205-hgp100. Each dot represents one single cell. A total of 9 clusters were identified and color-coded as indicated. UMAP visualization of single cells from combined Pmel-1 cells (left), Pmel-1 cells from WT mice (middle), and Pmel-1 cells from CD11c-β-catenin^active^ mice (right). (**B**) Distribution of Pmel-1 cells from either WT or CD11c-β-catenin^active^ mice within each of the 9 clusters as depicted in (**A**). (**C**) Bubble plots depicting the expression of top DEGs for UMAP clusters are shown in (**A**). (**D**) Violin plot visualizing expression of the top 3 DGEs associated with clusters 2, 3, and 4 among the UMAP clusters. (**E**) UMAPs depicting the module score of gene sets associated with effector or memory, and expression of selected gene markers for CD8 effector (*Gzmb*, *Ifng*, *and Prf1*) and memory (*Sell*, *Tcf7* and *Id3*) cells. UMAPs for Pmel-1 cells from WT (upper UMAPs) and CD11c-β-catenin^active^ mice (lower UMAPs) are shown. Gradient expression levels are color-coded as indicated. (**F**) GO enrichment analysis identifies multiple GO Biological Processes (BP), Molecular Function (MF), and Cellular Component (CC) that are downregulated in Pmel-1 cells primed in CD11c-β-catenin^active^ mice.

**Figure 4 vaccines-12-00460-f004:**
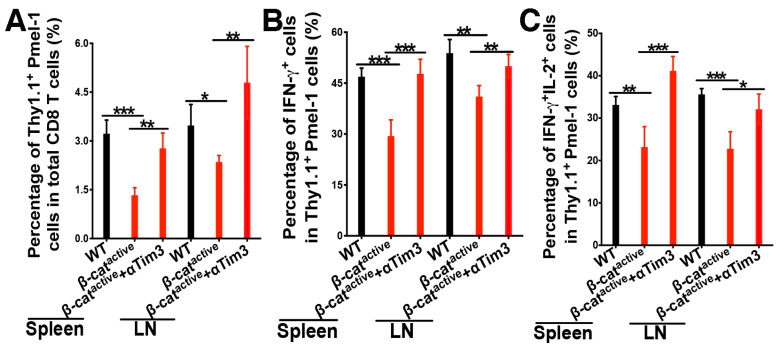
Anti-Tim-3 treatment reverses β-catenin-mediated inhibition of cross-priming. WT and CD11c-β-catenin^active^ mice were immunized with anti-DEC-205-hgp100 plus CpG, with half of the CD11c-β-catenin^active^ mice (*n* = 5) also being treated with anti-Tim-3 antibody. Cross-priming of adoptively transferred Thy1.1^+^ Pmel-1 CD8 T cells was examined on day 5 after immunization. (**A**) The percentages of Thy1.1^+^ Pmel-1 cells in total CD8 T cells (95% intervals: spleen, WT 3.2 ± 0.4%, CD11c-β-catenin^active^ 1.3 ± 0.2%, CD11c-β-catenin^active^ + αTim3 2.8 ± 0.5%; LN, 3.5 ± 0.6%, 2.4 ± 0.2%, 4.8 ± 1.1%), (**B**) the percentages of IFN-γ^+^ effectors out of total Thy1.1^+^ Pmel-1 cells (95% intervals: spleen, 46.9 ± 2.6%, 29.4 ± 4.8%, 47.7 ± 4.3%; LN, 53.8 ± 4%, 41.4 ± 3.2%, 50 ± 3.4%), and (**C**) the percentages of IFN-γ^+^IL-2^+^ effectors out of total Thy1.1^+^ Pmel-1 cells (95% intervals: spleen, 33.1 ± 2%, 23.2 ± 4.8%, 41.2 ± 3.4%; LN, 35.6 ± 1.4%, 22.8 ± 4%, 32.1 ± 3.6%) are shown. Data are representative of two experiments. One-way ANOVA and post hoc T-tests with Bonferroni correction were used. *** *p* < 0.001, ** *p* < 0.01, and * *p* < 0.05.

**Figure 5 vaccines-12-00460-f005:**
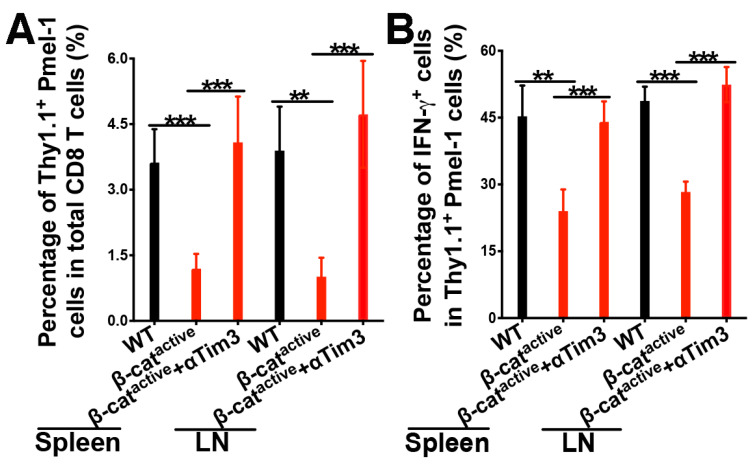
Anti-Tim-3 treatment reverses β-catenin-mediated inhibition of memory CD8 T cell responses. WT and CD11c-β-catenin^active^ mice were immunized with anti-DEC-205-hgp100 plus CpG, with half of the CD11c-β-catenin^active^ mice (*n* = 5) also being treated with anti-Tim-3 antibody. Recalled memory responses of adoptively transferred gp100-specific CD8 T (Pmel-1) cells were examined 5 days after recall with hgp100_25–33_ peptide on day 21. (**A**) The percentages of Thy1.1^+^Pmel-1 cells in total CD8 T cells (95% intervals: spleen, 3.6 ± 0.8%, 1.2 ± 0.3%, 4.1 ± 1%; LN, 3.9 ± 1%, 1 ± 0.4%, 4.7 ± 1.2%), (**B**) the percentages of IFN-γ^+^ effectors out of total Thy1.1^+^ Pmel-1 cells (95% intervals: spleen, 45.3 ± 6.9%, 24 ± 4.9%, 44 ± 4.6%; LN, 48.7 ± 3.2%, 28.3 ± 2.3%, 52.4 ± 4%) following a 5-hour in vitro stimulation are shown. Data are representative of two experiments. One-way ANOVA and post hoc T-tests with Bonferroni correction were used. *** *p* < 0.001, ** *p* < 0.01.

**Figure 6 vaccines-12-00460-f006:**
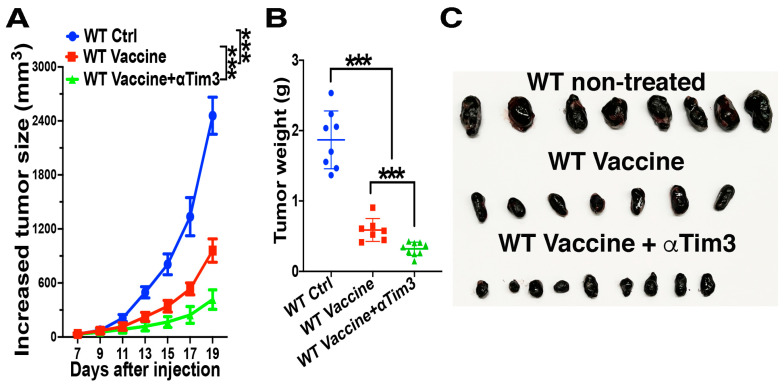
Anti-Tim-3 treatment improves the anti-tumor efficacy of DC-targeted vaccines using anti-DEC-205-hgp100. B16F10-bearing WT mice (*n* = 7–9) were not treated or were immunized with anti-DEC-205-hgp100 plus CpG with or without anti-Tim-3 treatment (tumor sizes around 3–6 mm), following Thy1.1^+^Pmel-1 CD8 T cell transfer. (**A**,**B**) Anti-Tim-3 treatment improves the anti-tumor efficacy of DC vaccines. Increased tumor sizes from the day of treatment are shown in (**A**) and tumor weight on day 19 in (**B**). A linear mixed model (Lme4) was fitted to the data in (**A**), and ANOVA for the fitted linear mixed model was then performed to determine the difference between groups. One-way ANOVA and post hoc T-tests with Bonferroni correction were used for (**B**). *** *p* < 0.001. (**C**) Photo of the tumors on day 19 after tumor inoculation. Data are representative of two experiments.

## Data Availability

The data that support the findings of this study are included in this manuscript and/or are available upon request by qualified researchers from the corresponding authors. scRNA-seq data have been deposited in GEO (accession number GSE263715).

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
