# Peer review of "β-Catenin in Dendritic Cells Negatively Regulates CD8 T Cell Immune Responses through the Immune Checkpoint Molecule Tim-3"

_vaccines, 2024, doi:10.3390/vaccines12050460_

Round 1

Reviewer 1 Report

Comments and Suggestions for Authors

The manuscript by Fu et al. investigates the role of β-catenin in dendritic cells (DCs) for the negative regulation of CD8 T cell responses in mice. The authors found higher levels of Tim-3 in DCs from CD11c-β-cateninactivemice. In subsequent experiments, the authors demonstrated an impaired prime/boost of specific CD8 T cell responses when using DEC-205 vaccination in CD11c-β- cateninactive mice. Single-cell RNA-seq data on specific CD8 T cells from DEC-205 vaccinated CD11c-β- cateninactive mice revealed a negative effect on genes related to cytotoxic capacity and memory development. Furthermore, the authors showed that a Tim-3-blocking antibody reversed the negative effect of β-catenin in DEC-205 vaccinated CD11c-β- cateninactivemice. Finally, the authors demonstrated that the efficacy of DEC-205 vaccination could be improved with Tim-3 blockade.

The aim of the manuscript addresses a relevant topic as the negative effect of β-catenin in DCs on the induction of CD8 T cell responses is not fully understood. Overall, the manuscript is well-written.

However, I am not convinced about a direct connection between β-catenin and Tim-3 in the negative regulation of CD8 T cell responses in CD11c-β-cateninactive mice. Although the authors showed a significant upregulation of Tim-3 on DCs in CD11c-β- cateninactive mice, the changes in mean fluorescence intensity (MFI) are minimal (MFI from 600 to 900, approximately 1.5 times higher MFI). On the other hand, the authors somehow ignored the significant down-regulation of PDL1 and up-regulation of PDL2 (MFI from 2000 to 2500, approximately 1.25 times higher MFI) on cDC1s from CD11c-β-catenin inactive mice.

Furthermore, DEC-205 targeting of antigens is known to induce tolerance if not using adjuvants, and therefore, the authors correctly performed DEC-205 vaccination in the presence of CpG to activate DCs. Hence, it should be excluded whether CpG acts similarly in β-cateninactive DCs compared to wildtype DCs. If not, a reduced adjuvant effect could also explain impaired CD8 T cell prime and boost. From this perspective, Tim-3 blockade (given its expression on CD8 T cells as well) might only compensate for impaired activation of DCs in β- cateninactive mice after DEC-205 vaccination.

Finally, the last experiment does not support the concept of a β-catenin/Tim-3 axis in CD8 T cell regulation by DCs, as Tim-3 treatment alone is missing.

Minor points:

·      Please provide an animal experimentation license number approving experiments performed in the study.

·      Please provide information about antibody clones and fluorochromes used.

·      Clarify what the error bars represent in figures (standard deviation, standard error of the mean, or confidence intervals). Adding 95% confidence intervals would facilitate the estimation of significances.

·      Improve the visibility of text, especially in part C of Figure 3.

·      As the data are derived from individual animals, figures with dots instead of bars would be more appropriate to visualize variances.         

Author Response

Reviewer #1

(1) “The aim of the manuscript addresses a relevant topic as the negative effect of β-catenin in DCs on the induction of CD8 T cell responses is not fully understood. Overall, the manuscript is well-written.”

Responses: Thank you for your encouraging comments on the manuscript. 

(2) “However, I am not convinced about a direct connection between β-catenin and Tim-3 in the negative regulation of CD8 T cell responses in CD11c-β-cateninactive mice. Although the authors showed a significant upregulation of Tim-3 on DCs in CD11c-β- cateninactive mice, the changes in mean fluorescence intensity (MFI) are minimal (MFI from 600 to 900, approximately 1.5 times higher MFI). On the other hand, the authors somehow ignored the significant down-regulation of PDL1 and up-regulation of PDL2 (MFI from 2000 to 2500, approximately 1.25 times higher MFI) on cDC1s from CD11c-β-catenin inactive mice.

Furthermore, DEC-205 targeting of antigens is known to induce tolerance if not using adjuvants, and therefore, the authors correctly performed DEC-205 vaccination in the presence of CpG to activate DCs. Hence, it should be excluded whether CpG acts similarly in β-cateninactive DCs compared to wildtype DCs. If not, a reduced adjuvant effect could also explain impaired CD8 T cell prime and boost. From this perspective, Tim-3 blockade (given its expression on CD8 T cells as well) might only compensate for impaired activation of DCs in β- cateninactive mice after DEC-205 vaccination.” 

Responses: We appreciate the insightful comments. We have modified the conclusions, and added discussion on PD-L1/PD-L2 as well as the effects of CpG (adjuvant) on WT and β-cateninactive DCs.

(3) “Finally, the last experiment does not support the concept of a β-catenin/Tim-3 axis in CD8 T cell regulation by DCs, as Tim-3 treatment alone is missing.” 

Responses: Thank you for your insightful comment regarding the last experiment. We did not include anti-Tim-3 treatment alone in this experiment due to previous studies indicating limited or no anti-tumor efficacy of anti-Tim-3 alone in the B16 melanoma model (e.g., SakuiShi K et al., 2010, JEM; Baghdadi M. et al., 2012, Cancer Immunol Immunother; Wenthe J. et al., 2022. Molecular Therapy Oncolytics; Luo J. et al., 2023. Theranostics).

Our study demonstrates that β-catenin upregulates Tim-3 in cDC1s (Fig 1), and combination of anti-Tim-3 treatment with cDC1-targeted vaccine reverse DC β-catenin-mediated inhibition of CD8 T cell responses (Fig 4-5) and anti-tumor immunity (Fig 6). Notably, Tim-3 on DCs instead of T cells mediates the anti-tumor efficacy of Tim-3 blockade (Dixon KO et al., 2021, Nature). Therefore, our findings suggest that a β-catenin/Tim-3 axis might serve as a potential mechanism to regulate DC vaccine-induced CD8 T cell responses. We have revised our statements in the manuscript accordingly.

(4) “Please provide an animal experimentation license number approving experiments performed in the study.” 

Responses: We have added a new Institutional Review Board Statement on the use of animal experiments with the Animal Protocol Number and Animal Welfare Assurance Number.

(5) “Please provide information about antibody clones and fluorochromes used.”

Responses: We thank the reviewer for pointing out this issue.  We have now included the information on antibody clones and fluorochromes used.

(6) “Clarify what the error bars represent in figures (standard deviation, standard error of the mean, or confidence intervals). Adding 95% confidence intervals would facilitate the estimation of significances.

As the data are derived from individual animals, figures with dots instead of bars would be more appropriate to visualize variances.”

Responses: We appreciate the reviewer's feedback on figure presentation. We have taken the following steps to address this concern: 1) clarified the representation of error bars in the Methods section; 2) provided 95% confidence intervals in Figure legends to aid in estimating variation and significance.

While we acknowledge the merits of dot plots for visualizing variances, bar graphs are commonly used, particularly for graphs involving derived numbers like percentages (graphs with dots are used in Fig 6B for tumor weight). We have also included 95% confidence intervals to enhance the information available for assessing variation and significance, as suggested by the reviewer. We appreciate your understanding in retaining the bar graphs for consistency and their widespread acceptance.

(7) “Improve the visibility of text, especially in part C of Figure 3".

Responses: Thank you for pointing this out. We have revised Figure 3 to enhance the visibility of text in Part C.

Reviewer 2 Report

Comments and Suggestions for Authors

Fu et. al. present a very well written manuscript describing a unique role for b-catenin in the function of dendritic cells. Specifically, they demonstrate that expression of a constitutively active b-catenin in dendritic cells decreases the ability of DC to activate transferred tumor specific CD8 T-cells. They then show that activation of b-catenin leads to the upregulation of the inhibitory molecule Tim-3 on DC, providing novel insight and an extension of their previous work. Although the results demonstrating reduced activation of T cells is rather convincing and confirms their previous findings it is not clear that the upregulation of Tim-3 is the only difference observed in b-catenin-active DC. A more thorough characterization of these DC and their function would greatly improve the manuscript. The following concerns should be addressed prior to publication. 

1. It is known that the CD11c-cre transgenic mouse can express cre in other lymphocytes subsets (PMID 38101508; DOI: 10.1016/j.jim.2023.113600). I would like to see a more in-depth characterization of other immune cell subsets and their expression of beta-catenin to rule out the possibility that overexpression of b-catenin is occurring in other subsets. Ideally, a dendritic cell transfer into a wild-type mouse would help to strengthen the evidence that this is a dendritic cell specific finding. 

2. The authors consistently use a vaccine approach with anti-DEC-205 plus CpG to demonstrate the role of b-catenin in DC, however this is only a single pathway for activating CD8 T-cells through DC. The authors should demonstrate that b-catenin in DC negatively regulates CD8 T cell responses using more than one model system in order to make this broad conclusion. 

3. Since wild-type mice express Tim-3 at a detectable level it would be critical to have control wild-type vaccinated anti-Tim3 treated mice in Figures 4 and 5 to differentiate the effect of anti-Tim-3 due to overexpression of b-catenin from native expression.   

4.  In Figure 6 the color coding of figure B appears to be different from that in figure A with the increased tumor volume not corresponding to the tumor weight. Is this correct? 

Comments on the Quality of English Language

No issues

Author Response

Reviewer #2

(1)Fu et. al. present a very well written manuscript describing a unique role for b-catenin in the function of dendritic cells. Specifically, they demonstrate that expression of a constitutively active b-catenin in dendritic cells decreases the ability of DC to activate transferred tumor specific CD8 T-cells. They then show that activation of b-catenin leads to the upregulation of the inhibitory molecule Tim-3 on DC, providing novel insight and an extension of their previous work. 

Responses: Thank you for your encouraging comments on the manuscript. 

(2) “Although the results demonstrating reduced activation of T cells is rather convincing and confirms their previous findings it is not clear that the upregulation of Tim-3 is the only difference observed in b-catenin-active DC. A more thorough characterization of these DC and their function would greatly improve the manuscript.” 

Responses: We appreciate the insightful comments and agree with the reviewer's point. In addition to Tim-3, our study in Supplementary Figure 2 demonstrates that WT and β-cateninactive DCs also differ in the expression in other immune checkpoint molecules such as PD-L1 and PD-L2. We have now included a discussion on PD-L1/PD-L2 to highlight that other inhibitory molecules besides Tim-3 could also play a role.

While our manuscript primarily focuses on immune checkpoint molecules, previous reports by us and others have investigated the function of β-cateninactive DCs in various contexts, including DC-targeted vaccine (Liang X et al. 2014 JLB. PMID 24023259; Fu C et al. 2015 PNAS. PMID 25730849), Th1/Th17 differentiation and autoimmune neuroinflammation (Suryawanshi A et al. 2015 JI. PMID 25710911), Vitamin A metabolism and regulatory T cell responses  (Hong Y et al., 2015 Cancer Res. PMID 25568183),  DC development (Cohen et al., 2015 JI. PMID 25416805), and DC maturation/function (Fu C et al. 2015 PNAS).

(3) “It is known that the CD11c-cre transgenic mouse can express cre in other lymphocytes subsets (PMID 38101508; DOI: 10.1016/j.jim.2023.113600). I would like to see a more in-depth characterization of other immune cell subsets and their expression of beta-catenin to rule out the possibility that overexpression of b-catenin is occurring in other subsets. Ideally, a dendritic cell transfer into a wild-type mouse would help to strengthen the evidence that this is a dendritic cell specific finding.” 

Responses: Thank you for your insightful comment regarding CD11c-cre transgenic mice. Using CD11c-cre-β-catenindel/del mice, Manicassamy et al. (2010, Science. PMID 20705860) extensively examined the effects of CD11c-cre on other immune cell subsets besides DCs, demonstrating specificity to DCs with minimal impact on macrophages, B cells, NK cells, and T cells in the spleen, with only a slight effect on intestinal macrophages. As CD11c-cre-β-cateninactive (CD11c-cre-β-cateninexon3/exon3) mice use the same CD11c-cre to delete exon 3 of b-catenin, we expect the same pattern as the conditional deletion mice.

Furthermore, Cohen et al. (2015, JI. PMID 25416805) demonstrated that β-catenin is specifically expressed in CD11c+ DCs but not in T cells in CD11c-cre-β-cateninactive mice. These findings strongly support the specificity of β-catenin expression in DCs and alleviate concerns about overexpression in other immune cell subsets. We have included these references to strengthen the discussion on the specificity of β-catenin expression in DCs in our manuscript.

(4) “The authors consistently use a vaccine approach with anti-DEC-205 plus CpG to demonstrate the role of b-catenin in DC, however this is only a single pathway for activating CD8 T-cells through DC. The authors should demonstrate that b-catenin in DC negatively regulates CD8 T cell responses using more than one model system in order to make this broad conclusion.” 

Responses: We appreciate the insightful comment and agree with the reviewer that our studies using anti-DEC-205 plus CpG should not lead to broad conclusions. We have included a discussion in our manuscript acknowledging the limitation of our current study.

It's important to note that previous studies by us and others, such as Liang X et al. (2014, JLB. PMID 24023259) and Cohen et al. (2015, JI. PMID 25416805), have shown that adjuvants or infections can regulate β-cateninactive DCs' function in CD8 T cell responses differently. These findings highlight the need for multiple model systems to fully understand the role of β-catenin in DC-mediated CD8 T cell responses. We are committed to expanding our research using additional model systems in future studies.

(5) “Since wild-type mice express Tim-3 at a detectable level it would be critical to have control wild-type vaccinated anti-Tim3 treated mice in Figures 4 and 5 to differentiate the effect of anti-Tim-3 due to overexpression of b-catenin from native expression.”

Responses: Thank you for your insightful comment. While we recognize that including control WT mice treated with anti-Tim-3 in Figures 4 and 5 could provide additional insights into the role of Tim-3 expression in WT DCs, our focus in this study was on investigating the potential interaction between β-catenin and Tim-3.

Our findings in Figures 4 and 5 demonstrate that combining anti-Tim-3 treatment with a cDC1-targeted vaccine restores cross-priming and memory responses of gp100-specific CD8 T cell responses in CD11c-β-cateninactive mice. This suggests that anti-Tim-3 treatment synergizes with DC vaccines to reverse a β-catenin-mediated inhibition of CD8 T cell responses, which has been shown in this manuscript and previous studies (Liang X et al. 2014 JLB. PMID 24023259; Fu C et al. 2015 PNAS. PMID 25730849). Furthermore, as shown in Figure 1, β-catenin upregulates Tim-3 in cDC1s. These results provide support for a potential β-catenin/Tim-3 axis as a mechanism to regulate DC vaccine-induced CD8 T cell responses. However, we also acknowledge that other upregulated inhibitory molecules such as PD-L2 might also play a role, and we have revised our manuscript to reflect this important point.

(6) “In Figure 6 the color coding of figure B appears to be different from that in figure A with the increased tumor volume not corresponding to the tumor weight. Is this correct?”

 Responses: We thank the reviewer on pointing out this mistake.  We have now corrected the error.

Reviewer 3 Report

Comments and Suggestions for Authors

Reviewer comments and suggestions

The authors in this study reported that activation of b-catenin leads to the up-regulation of inhibitory molecule T-cell immunoglobulin and mucin domain 3 (Tim-3) in type 1 conventional DCs (cDC1s). They have used a cDC1-targeted vaccine model with anti-DEC-205 engineered to express the melanoma antigen--human gp100 (anti-DEC-205-hgp100) and explored a CD11c-b-catenin active mice exhibited impaired cross-priming and memory responses of gp100-specific CD8 T (Pmel-1) cells upon immunization with anti-DEC-205-hgp100. Additionally the study showed that treating CD11c-b-catenin active mice with anti-Tim-3 antibody upon anti-DEC-205-hgp100 vaccination led to restored cross-priming and memory responses of gp100-specific CD8 T cells. Combining anti-DEC-205-hgp100 vaccines with anti-Tim-3 treatment significantly reduced tumor growth in B16F10-bearing mice, compared to DC vaccine alone.

Overall, the manuscript was written nice. However, a few major concerns/comments needed to be explained or modified

  1. Lines 45 These references should be discussed; otherwise, there is no importance of citing many
  2. Line 47 Similar comments as above
  3. Line 5456 The lines need to be explained well for the common reader of your paper
  4. Line 74-79 Please avoid long sentences
  5. Line 80-83 These points seem to be in result and discussion section. Please read from line 80-88 again and try to hypothesis your study first not to elaborate your findings at the end of the introduction section 
  6. Line 99-100 You need to mention the protocol number or ethical clearance number
  7. Line 181 is that 2 was a typo error, please check and also 323 
  8. Line 235 and several places I observed that the authors explained his previous publications
  9. Line 442-443 No need to add up references here at least the authors have to describe all their findings in the first paragraph of the discussion
  10. Line 445-447 This was your study results so I think it is not required to cite for any references; it would be better to discuss these results elsewhere.
  11. Line 490-496 and other places Please discuss your results by inserting the figure and table number for easy comprehension.
  12. Please check the reference 5

Author Response

Reviewer #2

(1) “Overall, the manuscript was written nice.”

Responses: We thank the reviewer for your encouraging comments on the manuscript. 

(2) “Lines 45 These references should be discussed; otherwise, there is no importance of citing many,

Line 47 Similar comments as above” 

Responses: We agree with the reviewer on this point. We have reduced the number of references to include only the most recent and relevant ones.

(3) “Line 54-56 The lines need to be explained well for the common reader of your paper,

Line 74-79 Please avoid long sentences.”

Responses: Thank you for highlighting these areas. We have rewritten these paragraphs to improve clarity and conciseness for the readers.

(4) “Line 80-83 These points seem to be in result and discussion section. Please read from line 80-88 again and try to hypothesis your study first not to elaborate your findings at the end of the introduction section.” 

Responses: We would like to thank the reviewer for your insightful comments. We have now moved the paragraph to the new "Conclusion" section, and rewritten a new paragraph to state our study Aims/hypothesis in the Introduction section.

(5) “Line 99-100 You need to mention the protocol number or ethical clearance number”

Responses: We have added the Animal Protocol Number and Animal Welfare Assurance Number in Institutional Review Board Statement.

(6) “Line 181 is that 2 was a typo error, please check and also 323.” 

Responses: We have reviewed the manuscript and corrected these errors.

(7) “Line 235 and several places I observed that the authors explained his previous publications."

Responses: We agree with the reviewer on this point. We have now removed these paragraphs from the Results section.

(8) “Line 442-443 No need to add up references here at least the authors have to describe all their findings in the first paragraph of the discussion.

Line 445-447 This was your study results so I think it is not required to cite for any references; it would be better to discuss these results elsewhere."

Responses: We thank the reviewer for this point. We have now revised the first paragraph of the Discussion section to describe our findings and removed the citations as you suggested.

(9) “Line 490-496 and other places Please discuss your results by inserting the figure and table number for easy comprehension."

Responses: We have added the Figure numbers in Discussion section.

(10) “Please check the reference 5."

Responses: The incorrect citation has been removed. Thank you for bringing it to our attention.

Reviewer 4 Report

Comments and Suggestions for Authors

Comments to authors:

The authors should add graphic abstract.

The authors would do English editing.

Introduction:

Lines 74 to 88: please remove it to the result section and rewrite the aim of the study at the end of introduction part.

Methods:

Mice and treatment

The authors should mention the ethical approval number regarding the mice part.

Results:

Β-catenin upregulates the expression of inhibitory molecule Tim-3 in DCs: lines from 191 to 200: this part more suitable in the discussion part as you compare your results with previous ones.

β-catenin in DCs suppresses tumor antigen-specific CD8 T cell responses: lines from 234 to 240: this section could be fitting in the discussion part.

Lines from 356 to 359: this part could be moved to conclusion part.

Discussion:

The authors start the discussion with “we identified/we demonstrated” and then add reference. Please clarify if you compare these results with previous ones as you mentioned “this was in consistent with previous studies …….et al. reported …...

Please if you compare these results with your previous one, you can mention “our previous report or study” to make it clear for the readers.

References:

References 1&2; please try to cite a recent ones.

Comments on the Quality of English Language

The authors would do English editing.

Author Response

Reviewer #3

(1) “The authors should add graphic abstract.”

Responses: We thank the reviewer for this excellent suggestion. We have now included a graphic abstract.

(2) “The authors should mention the ethical approval number regarding the mice part.” 

Responses: We have added the Animal Protocol Number and Animal Welfare Assurance Number in Institutional Review Board Statement.

(3) “Β-catenin upregulates the expression of inhibitory molecule Tim-3 in DCs: lines from 191 to 200: this part more suitable in the discussion part as you compare your results with previous ones.

β-catenin in DCs suppresses tumor antigen-specific CD8 T cell responses: lines from 234 to 240: this section could be fitting in the discussion part.

Lines from 356 to 359: this part could be moved to conclusion part.”

Responses: We appreciate your insightful comments and have revised these sections accordingly. We either removed these paragraphs or relocated them to the appropriate sections in the discussion or conclusion.

(4) “The authors start the discussion with “we identified/we demonstrated” and then add reference. Please clarify if you compare these results with previous ones as you mentioned “this was in consistent with previous studies …….et al. reported …...

Please if you compare these results with your previous one, you can mention “our previous report or study” to make it clear for the readers.” 

Responses: We agree with the reviewer on this point. We have now revised the Discussion section as you suggested.

(5) “References 1&2; please try to cite a recent ones.”

Responses: References 1 and 2 have been removed, and we have updated the citations with more recent references.

Round 2

Reviewer 1 Report

Comments and Suggestions for Authors

The authors satisfactorily addressed all of my concerns. They have made changes in the manuscript, including addressing potential limitations of their study, and overall, these changes have significantly improved the quality of the manuscript.

Reviewer 2 Report

Comments and Suggestions for Authors

Although I would like to see additional experiments performed to address my concerns, the authors have sufficiently addressed my concerns by addressing the limitations of the study.